# Increased brain size of the dwarf Channel Island fox (*Urocyon littoralis*) challenges "Island Syndrome" and suggests little evidence of domestication

Kimberly A. Schoenberger[1,2]*, Xiaoming Wang[2], Suzanne Edmands[1]

**1** Department of Biological Sciences, University of Southern California, Los Angeles, California, United States of America, **2** Department of Vertebrate Paleontology, Natural History Museum of Los Angeles County, Los Angeles, California, United States of America

* kschoenb@usc.edu

## Abstract

Although changes in overall body size during species' island adaptation is a well-established phenomenon, there are mixed findings regarding how brain size changes within resource-limited insular environments. Work on this issue has focused on fossil species and herbivores, with limited studies on carnivores and extant island species. Here, we aim to close this knowledge gap and expand our understanding of brain size evolution by examining the relative brain size of the extant island canid, the Channel Island fox (*Urocyon littoralis*) amongst its six island-specific subspecies and in comparison to its larger mainland relative, the gray fox (*Urocyon cinereoargenteus*). As the island fox was likely brought to the southern Channel Islands by indigenous peoples, this research is also relevant in exploring the impact of human transport and potential domestication on brain size. Our endocranial analysis found that foxes across five of the islands have a moderately higher relative brain size in comparison to the gray fox, with only the second smallest, most geographically isolated island, San Nicolas, exhibiting reduction. No significant differences in encephalization were found between sexes within any subspecies. These findings suggest that the selective pressures driving reduced body size on islands may not outweigh the adaptive benefits of increased brain size, with the exception of highly resource-constrained environments such as on San Nicolas. Disparity in brain size among the three southern islands and the increased encephalization of San Clemente and Santa Catalina foxes compared to the mainland gray fox further suggests that although humans may have facilitated transport of the southern island foxes, true domestication was likely not practiced. Broadly, this research indicates that brain size reduction is not a straightforward trait of island adaptation, and changes in insular species' brain size will vary in conjunction with island-specific selective pressures.

**Data availability statement:** All raw specimen data, associated R code, and supplementary figures are included in the manuscript and its Supporting information files. All 3D image files (CT scans and mesh PLYs) for brain endocasts are available from Morphosource at www.morphosource.org/projects/000739635 (Project ID: 000739635). Specific DOIs as follows: island fox endocast (https://doi.org/10.17602/M2/M740219), gray fox endocast (https://doi.org/10.17602/M2/M740195), island fox raw CT data (https://doi.org/10.17602/M2/M740169), gray fox raw CT data (https://doi.org/10.17602/M2/M740136).

**Funding:** Funding for this project and Kimberly's PhD research was provided by Dornsife College of Letters, Arts and Sciences at the University of Southern California; the Wrigley Institute for Environmental Studies and Offield Family Foundation; and the USC Women in Science and Engineering Graduate Fellowship.

**Competing interests:** The authors have declared that no competing interests exist.

## Introduction

### Island encephalization

How brain size in mammal species changes in response to available resources, behavior, predation, and other selective stressors is a long-contended field of research, particularly as it pertains to changes during island adaptation. To account for the variation in brain size among mammals, comparisons can be made through use of the Encephalization Quotient (EQ), the ratio of actual brain size relative to the expected brain size in a living mammal of the same body size [1]. In many instances, phyletic dwarfs on islands have been thought to have a substantially reduced EQ compared to their mainland counterparts [2–4]. These smaller "island brains" have been considered to be a foundational characteristic of "Island Syndrome", a phenomenon in which island-dwelling species develop a set of distinct morphological and behavioral characteristics including dwarfism and gigantism, simplified locomotion and coloration (typically found in insular birds, e.g., loss of flight and dulling of plumage), and increased tameness and sociality [5–9]. A reduced EQ being inherent to island adaptation has commonly been attributed to the tradeoff between the energetically high expense of brain tissue relative to an organism's body size versus the limited resources on islands as well as a reduction in predation pressures [2]. However, recent findings have shown this may not be as widely applicable as previously thought, with a growing number of species being found to demonstrate increased EQ during island adaptation despite decreasing their overall body size [10].

### Insularity in *Urocyon*

The island fox (*Urocyon littoralis*) provides an excellent opportunity to further these studies. It is a prime example of island dwarfing at two-thirds the size of the mainland gray fox (*Urocyon cinereoargenteus*) [11] and is endemic to six of the eight Channel Islands of California, each with a unique subspecies. Though all island fox subspecies exhibit dwarfism, average body size varies somewhat among the islands and does not appear to be associated with geographic area, as the smallest subspecies is found on the largest island (Santa Cruz) and the second largest subspecies is found on the smallest island (San Miguel) [12]. The island fox is also of particular interest as the world's only extant canid restricted entirely to islands; though there are other insular carnivore species, few are island-exclusive, and most are conspecific with mainland populations [13]. Further, both island and gray foxes exhibit minimal sexual size dimorphism (SSD), with males only marginally larger than females, and this difference is slightly greater in gray foxes [11,12,14]. This provides an interesting contrast to another island species, the dwarf elephant *Elephas falconeri*, which has been found to increase SSD during island adaptation, potentially as a result of higher levels of competition for females [15]. The combination of these factors in the island fox thus suggests that a range of selective pressures may be at play in regard to energy expenditure in the face of limited island resources.

In addition, the mainland gray fox, which gave rise to the island foxes, is widely available within 30–100 miles to the nearest shoreline of the islands, affording an

optimal comparison for an ancestral/sister species. It is important to note, however, that although the gray fox is widely distributed throughout North and Central America with 16 recognized subspecies [16], there is substantial divergence between Eastern and Western populations, which split approximately 0.8 million years ago [17]. Within this divide, the island fox forms a distinct cluster most closely related to populations west of the North American Continental Divide [17], particularly those in California [18]. As such, this study only utilized specimens from four Western gray fox subspecies, all which cluster genetically close to one another [17] and can be found in ranges along or within 150 kilometers of the Pacific Coast. We verified that the gray fox subspecies were not significantly different in key phenotypic measurements (body mass, skull length, encephalization; S1 Fig) and therefore combined them into a single group for further analyses and comparisons to the island fox subspecies.

The Channel Island fox is also not the only documented instance of insularity within *Urocyon*; skeletal remains of another diminutive gray fox relative have been documented on Cozumel Island off the coast of Eastern Mexico [19]. Unfortunately, the Cozumel Island fox has not been formally classified as a distinct species due to very limited material (a singular cranium and miscellaneous postcrania). However, the size of the skeletal remains indicates that individuals living on the island were 60–80% the size of the mainland gray fox, and authors suggest that colonization predates human arrival [19].

### Migratory history of the Channel Island fox

Genomic and radiocarbon evidence indicate the fox likely dispersed to the six Channel Islands in two waves—first, by natural migration to the three northern islands (then, the superisland of Santarosae when sea levels were higher), followed by a secondary dispersal to and among the three southern islands, likely through human transport [11,20,21]. The proposed natural migration to Santarosae is also supported by the findings in Cozumel [19], where foxes would have had to travel a similar distance from the mainland to colonize the island without the assistance of humans.

However, the second wave, human-assisted dispersal offers another avenue through which the island fox provides a unique test subject—examining potential effects of anthropogenic interactions on encephalization. Although the intensity of the relationship between indigenous peoples and island foxes is unclear, the distance of dispersal in combination with archaeological findings suggests some degree of cohabitation of island foxes and humans [20–22]. This has implications for encephalization because cohabitation and reliance on humans, with the ultimate form being domestication, has been repeatedly shown to drive a decrease in brain size and EQ [23–27]. This may be attributed to the significantly reduced cognitive function required for domesticated animals when food, shelter, and other necessities are readily available from human populations. As such, examining variation in EQ among the six island fox subspecies as well as in comparison to the mainland gray fox will help us further understand how mammalian neuroanatomy may evolve in the face of human interactions, geographic isolation, and limited resources.

Here, we explore brain size changes during the island dwarfing and insular evolution of the island fox, and how those changes may have contributed to its long-term survival on the Channel Islands. We performed measurements and analyses of brain size, cranial shape, and body size of the six island fox subspecies and four mainland gray fox subspecies, further subsetting by sex. The findings were then contrasted to other insular dwarf species to examine where EQ of the island fox falls in a broader context.

## Materials and methods

### Sampling

This study used existing skeletal specimens from mammalogy and vertebrate zoology collections at the Natural History Museum of Los Angeles County (LACM) and the Santa Barbra Museum of Natural History (SBNMH), respectively. As no new specimens were collected and no animals were trapped or sacrificed for the purposes of this work, no permits were

required and the described study complied with all relevant regulations. A total of 287 skulls were used in the data analysis distributed among the six island fox subspecies (*U. l. littoralis, U. l. santarosae, U. l. santacruzae, U. l. dickeyi, U. l. catalinae, U. l. clementae*) ([Table 1](), [Fig 1B]()) and four gray fox subspecies (*U. c. californicus, U. c. scottii, U. c. townsendi, U. c. nigrirostris*) ([Fig 1A]()). Only complete skulls with known metadata were used; skulls that were damaged to the point that key measurements could not be taken were excluded. This sample size is of particular significance for research into brain size of insular mammals, as existing studies rely on fossil specimens, often with very limited sample sizes (sometimes just a single specimen) [2–4,10,28]. The full list of specimens used in this study is available in [S1 Table]().

## Volumetric and linear measurements

Brain sizes for all specimens were measured using endocranial volume (ECV), where one cubic centimeter serves as a proxy for one gram of brain weight [1]. ECV measurements were obtained by pouring glass microbeads of approximately 1 mm in diameter into the foramen magnum of each skull and gently tapping and tamping down to ensure all cranial pockets were full. To prevent spillage from the endocranial cavity, other foramina were sealed with flexible putty prior to filling. The beads were then transferred to a graduated cylinder to obtain the cranial volume (±0.1 ml accuracy). These measurements were corroborated by taking volume measurements of digital endocasts from CT scans of 10 sample skulls (5 gray fox, 5 island fox) using Avizo Lite 2019.2 (Thermo Fisher Scientific). Endocasts were segmented from CT scans, in which a surface area volume of the endocast was generated and the Avizo built-in statistical module was used to extract the surface area volume of the model in cubic centimeters. Digital endocranial volumes were all within ±0.96 cm$^3$ of the bead-based volumetric measurements ([S2 Table]()). Endocast models were then compared to examine any key structural differences in the brain between the gray and island fox.

Linear measurements were also taken to examine impacts of external cranial shape variation on any brain size differences. Eleven cranial measurements ([Fig 2]()) were taken for each specimen using digital calipers (Mitutoyo, ±0.01 mm Accuracy). These specific measurements are standard practice in cranial morphometry and were used as a proxy to the key areas of shape variation in vertebrates [34].

## Body mass and relative brain size

Body masses (BM) of all specimens were estimated from a Carnivora-specific regression model utilizing measurements of the occipital condylar width (OCW), with the formula $\ln(BM)=8.5852*\ln(OCW)^{2/3}-10.2696$ [35]. This method is particularly useful in contrast to other proxies that use limb bone dimensions to estimate body mass [36], as many museum skull specimens do not have associated post-crania. Calculated body masses for all specimens fell within their normal ranges as measured from live specimens—between ~1–3 kg for island foxes [37] and ~3.5–7 kg for gray foxes [16]. EQ for all specimens was calculated from measured ECV and calculated body mass using the formula $EQ=ECV/0.12*BM^{2/3}$ [1]. For additional comparison, we also used a simpler percentage ratio of brain mass to body mass with the formula

**Table 1. Channel Islands inhabited by island fox subspecies.**

| Northern Islands | Subspecies | Total Area (km²) | Distance to closest land mass |
|---|---|---|---|
| San Miguel (SMI) | *U. l. littoralis* | 37.74 | 4.9 km to Santa Rosa |
| Santa Rosa (SRI) | *U. l. santarosae* | 215.27 | 4.9 km to San Miguel, 8.9 km to Santa Cruz |
| Santa Cruz (SCZ) | *U. l. santacruzae* | 249.95 | 8.9 km to Santa Rosa, 30.3 km to mainland |
| **Southern Islands** | | | |
| Santa Catalina (SCA) | *U. l. catalinae* | 194.19 | 33 km to mainland |
| San Clemente (SCI) | *U. l. clementae* | 147.13 | 34 km to Santa Catalina |
| San Nicolas (SNI) | *U. l. dickeyi* | 58.93 | 82 km to Santa Catalina, 79 km to Clemente |

**Fig 1. Map of island and gray fox subspecies specimens used in this study.** Map showing ranges of gray and island fox subspecies used in this study (map data from Natural Earth [29] and NOAA [30]) A: Map of North America, regions for gray fox subspecies specimens indicated by highlighted colors: *U. c. californicus* (dark yellow), *U. c. scottii* (light yellow), *U. c. townsendi* (light green), *U. c. nigrirostris* (dark green). Striped pattern for California illustrates presence of multiple subspecies. Data points show locations where specimens were collected: diamonds indicate specimens with exact coordinates, X-circles indicate specimens where only the general area was documented. U.S. state abbreviations: California (CA), Arizona (AZ), New Mexico (NM); Mexico province abbreviations: Sonoma (SO), Jalisco (JA), Colima (CL). Dashed blue line illustrates Continental Divide [31]. Dark gray rectangle indicates area represented in B. B: Inset map of the Channel Islands of California. Northern islands in blue shades (San Miguel (SMI), Santa Rosa (SRI), and Santa Cruz (SCZ)), southern islands in red shades (Santa Catalina (SCA), San Clemente (SCI), and San Nicolas (SNI)). Due to the small geographic range of the islands, exact collection locations for island fox specimens not shown. Smaller islands not inhabited by foxes shaded in dark gray (Anacapa (AI) and Santa Barbara (SBI)). Outline of light gray shade around northern islands indicates maximum shoreline of superisland Santarosae at glacial maxima [32], before the arrival of the island foxes. Santarosae shoreline generated from NOAA bathymetry data [33] with estimated sea level 120 m below present [32]. Blue arrow indicates first wave of migration to Santarosae, orange arrows indicate second wave of migration to Southern islands.

$\% = 100 * ECV/BM$. Standardizing with these equations accounts for the fact that the brain scales allometrically to body size and creates a normalized unit of comparison between differently sized species [1,38,39].

### Statistical analyses

Statistical analyses were conducted primarily in RStudio 2023.06.0 [40] with additional use of PAST (Paleontological statistics software) [41] where indicated. First, we examined differences in allometric scaling of brain size to body mass and brain size to skull length within *Urocyon* and in contrast to expected values from other carnivorans using linear regression models. Measurements were natural log transformed prior to regression tests and modeling. We then used ANCOVA, including interaction terms, to test whether slopes and intercepts of these scaling relationships differed significantly between species.

Second, we examined EQ values for each species and subspecies. Data were assessed for normality using Shapiro-Wilk tests and for homoscedasticity using Levene's test. Comparison groups that were not normally distributed were compared using Mann-Whitney tests of statistical significance. If test groups were normally distributed but did not have equal variance, a one-way ANOVA accounting for unequal variance was utilized to test for statistically significant differences

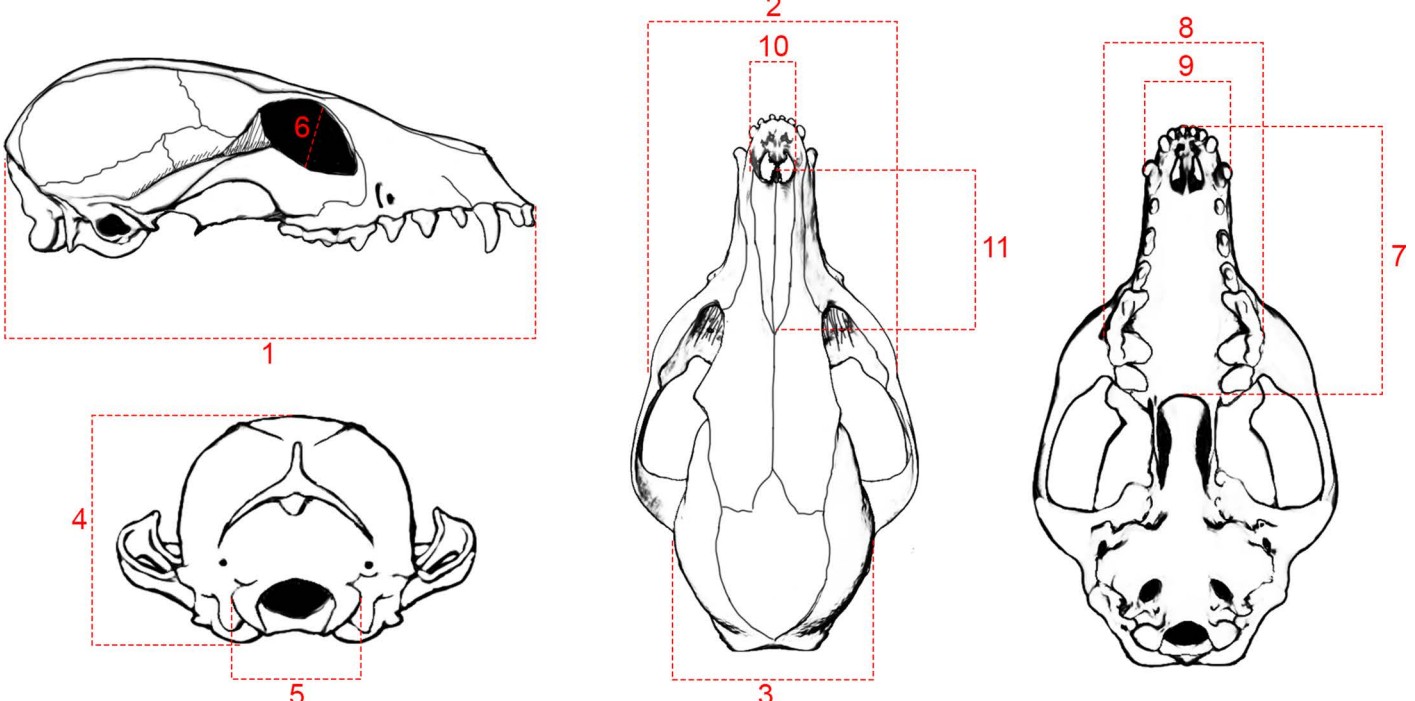

**Fig 2. Schematic diagram of 11 linear measurements taken on skulls.** 1: Total maximum skull length (TSL), 2: Zygomatic width (ZW), 3: Cranial vault width (CVW), 4: Cranial vault height (CVH), 5: Occipital condylar width (OCW), 6: Orbital diameter (OD), 7: Palatal length (PL), 8: Palatal width (PW), 9: Bicanine width (BCW), 10: Nasal width (NW), and 11: Nasal length (NL).

among groups. Once significance was established, Dunnett's T3 test was used post hoc for pairwise comparison of group means to an alpha level of 0.05, accounting for multiple comparisons. Results were visualized using box plots.

Finally, we performed principal component analysis (PCA) of linear measurements using PAST to further explore contributions of cranial shape variation to EQ values. Only adult specimens were used to mitigate potential impacts of ontogenetic shape differences. PCA using a variance-covariance matrix was conducted first on log-transformed raw linear measurements and then on log-transformed linear measurements normalized to a skull length of 1 to account for size variation among groups. Group differences were assessed using MANOVA on all principal components, followed by post-hoc Hotelling's tests with Bonferroni correction to identify significantly different pairings. PCA plots were generated in RStudio. If outliers were identified visually, they were removed from the dataset and PAST analyses were re-run prior to final result reporting.

## Results

### Brain scaling within *Urocyon*

Between species, the island fox presented a higher relative brain size compared to the mainland gray fox (Mann-Whitney test *p*-value: $5.12e-06$; Fig 3). Log-transformed regression of brain versus body mass of island and gray fox specimens placed all individuals in this study above the expected brain volume of caniform carnivorans by mass [39], but to a greater extent in the island fox than the gray (Fig 4A). Log-transformed regression of brain volume versus skull length showed similar scaling between species (Fig 4B). Slopes and intercepts did not differ significantly between species for either model (Table 2). Intraspecific encephalization values and slopes were consistent with encephalization patterns found in other canid studies, wherein linear regression slopes flatten within

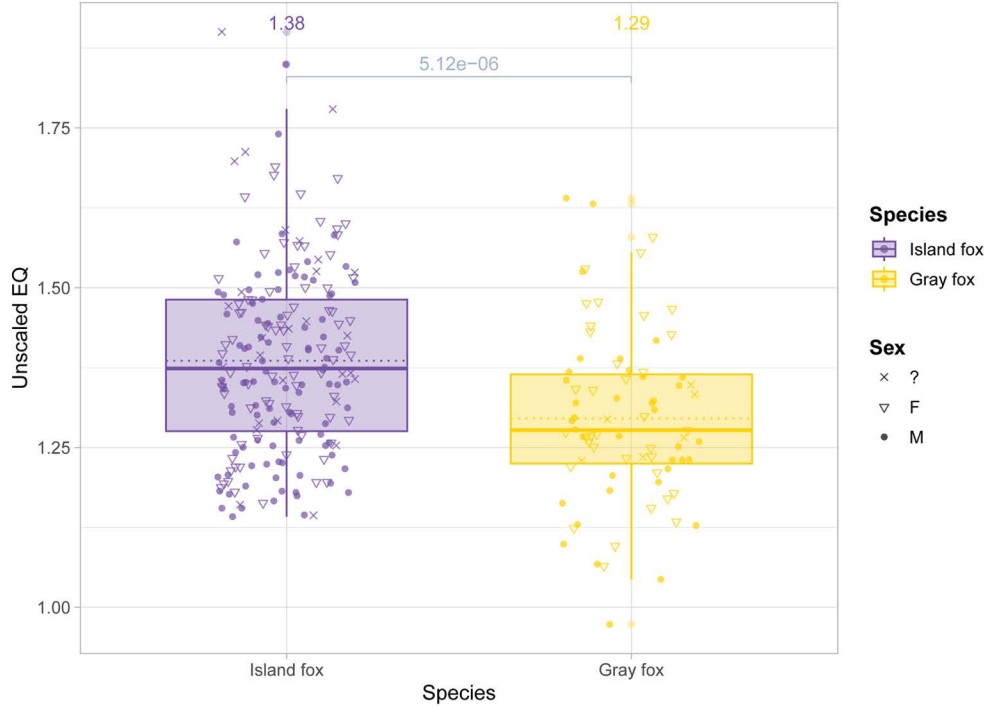

**Fig 3. Relative brain size (EQ) between island and gray fox species.** Mann-Whitney test *p*-value shown in brackets. Mean EQ noted above each species and denoted on plot by dotted line. Median EQ values denoted by solid line.

species-specific models and begin to run parallel to other species slopes, in contrast to the steeper slope of the over-all taxon model [24].

When tested for statistical differences in EQ, TSL, BBMR, and BM, the four gray fox subspecies did not differ significantly from one another (S1 Fig) and were thus grouped together as "gray fox" for clarity of EQ comparisons to the island fox subspecies. Per island fox subspecies, five out of six demonstrated significantly higher EQ compared to the gray fox, with only San Nicolas presenting a lower EQ (Fig 5). Brain-body mass ratio comparisons showed very similar patterns to EQ, with the exception of San Nicolas still presenting low values but not differing significantly from the gray fox (S2 Fig). The majority of the other islands had similar EQ values, and only Santa Cruz foxes showed both significantly higher EQ and brain-body mass percentage than the gray fox and other island fox subspecies. Further separation by sex did not present statistically significant differences in EQ within respective groups (S3 Fig).

PCA of linear skull measurements revealed that the gray fox and island fox clustered separately in analyses of both log-raw and log-normalized measurements (Fig 6). For log-raw PCA, PC1 and PC2 represented 87.78% of variance; for log-normalized PCA, PC1 and PC2 represented 69.40% of variance. PCA separating gray fox subspecies did not show significant differences between groups, and as above, were thus grouped together as "gray fox" for overall PCA comparisons. The island fox broadly clustered together, with some slight differences in clustering by subspecies (Fig 6A, C). Loadings for log-raw data showed differences were primarily driven by overall skull size along PC1 (Fig 6B), and loadings for log-normal data showed clustering driven mainly by snout size, particularly by nasal width (Fig 6D). MANOVA of all principal components using the gray fox species and island fox subspecies as coherent groups showed presence of significant differences between groups (log-raw data: Wilks' lambda=0.01342, Pillai trace=2.362, *p*<0.0001; log-norm data: Wilks' lambda=0.03066, Pillai trace=2.188, *p*<0.0001). Post-hoc Hotelling's test of principal components for both

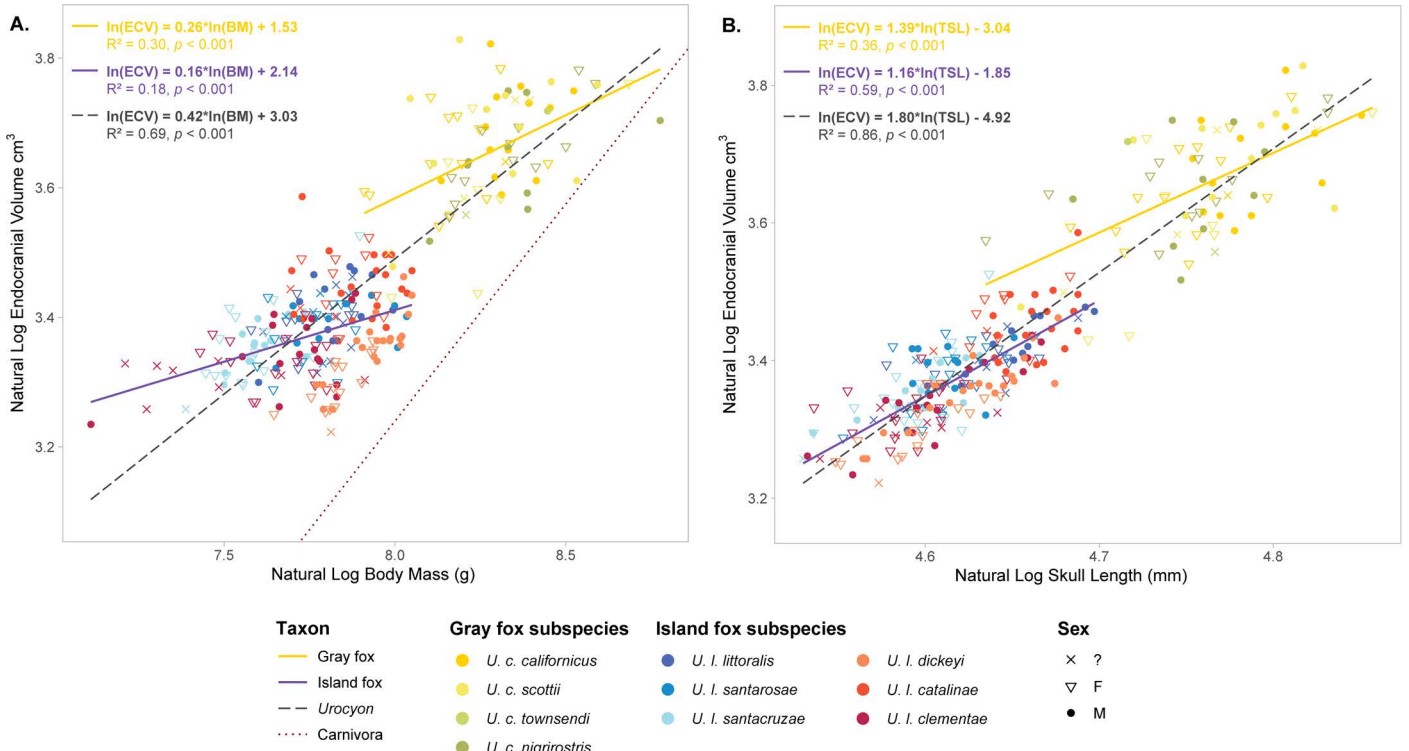

**Fig 4. Encephalization slopes of body mass versus brain mass (A) and skull length versus brain mass (B).** Sloped lines represent linear regression calculated for different groups: *Urocyon* (dashed black), island fox (solid purple), and gray fox (solid yellow). Dotted red line in A represents the expected brain volume of carnivora by mass (32). General equation, $R^2$, and *p*-value indicated in top left corner.

**Table 2. Slopes and intercepts for linear regression models.**

| Model value | Island fox | Gray fox | Difference | *p*-value |
|---|---|---|---|---|
| ln(BM)~ln(ECV) Slope | 0.16 | 0.26 | −0.10 | 0.06 |
| ln(BM)~ln(ECV) Intercept | 2.14 | 1.53 | 0.60 | 0.16 |
| ln(TSL)~ln(ECV) Slope | 1.39 | 1.16 | 0.23 | 0.16 |
| ln(TSL)~ln(ECV) Intercept | −3.04 | −1.85 | −1.20 | 0.13 |

datasets showed significant differences between almost all pairings, but the most significant differences were found between gray-island pairings (S3 Table). Though still significantly different, the most similar pairing in the log-raw dataset was Santa Rosa-San Miguel; from the log-normal dataset, Santa Cruz-Santa Rosa and Santa Cruz-Santa Catalina.

## Comparison of cerebral cortex regions and structures

When comparing the digital endocasts of the island and gray fox, the key difference we found was in the folding of the precruciate and postcruciate gyri, and the depth of the cruciate sulci (Fig 7). The island fox endocast exhibited slightly deeper cruciate sulci and larger gyri than the gray fox in this area (Fig 7E,J). The size and shape of the olfactory bulb appeared similar between the two species, although the island fox exhibited some reduced length in the prefrontal area.

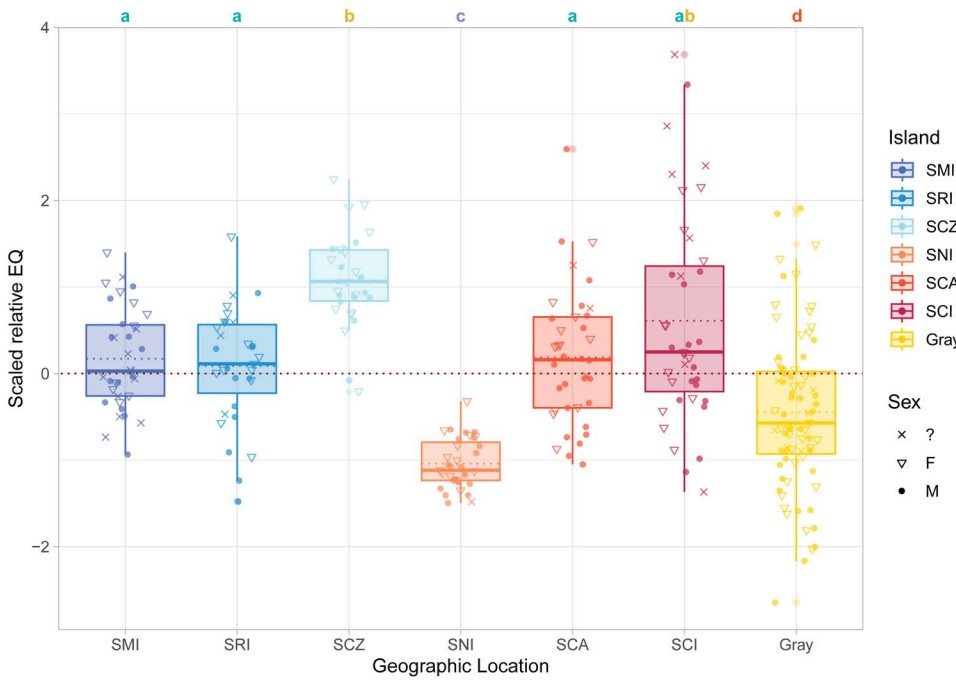

**Fig 5. Relative brain size (EQ) of gray fox and island fox subspecies.** Values scale normalized to zero. Means not sharing any letter are significantly different by Dunnett's T3 test at 5% level of significance.

### Brain scaling among other insular dwarfs

EQ of the island fox was found to be 0.09 higher than the gray fox on average (Table 3). The greatest change was demonstrated on Santa Cruz Island, with an EQ increase of +0.22 compared to the mainland. This value is similar to the dwarf fossil canid *Cynotherium*, which differed from its mainland relative by +0.30. Changes for each island subspecies in comparison to other insular species EQ can be found in Table 3.

### Discussion

Increasing brain size and complexity in Cenozoic mammals in general [1,38] and carnivorans in particular [44] compared to their ancestors has long been known. Long-term coevolution of predators and prey relationships was likely a major driver. Despite the apparent advantage of larger brain size for carnivorans, the brain as a tissue is energetically expensive [45] and increasing brain size therefore must be balanced with availability of resources. Island dwarfism, particularly for large vertebrates, is commonly interpreted as a result of resource limitations and/or reduced predatory pressure [46]. A relative reduction in brain size in island bovids helped to advance the notion that brain reduction can be an effective strategy in island forms [2]. As shown in Table 3, however, brain size increase or decrease in insular dwarfs is species specific. For small canids, the selective pressures driving small body size may also select for an increase in brain size, as seen in the island foxes.

### Drivers of larger relative brain size

With the single exception of San Nicolas foxes (see discussion below), the general rule seems to be a modest increase of EQ in Channel Island foxes. There are advantages to maintaining or evolving larger relative brain size, despite it being

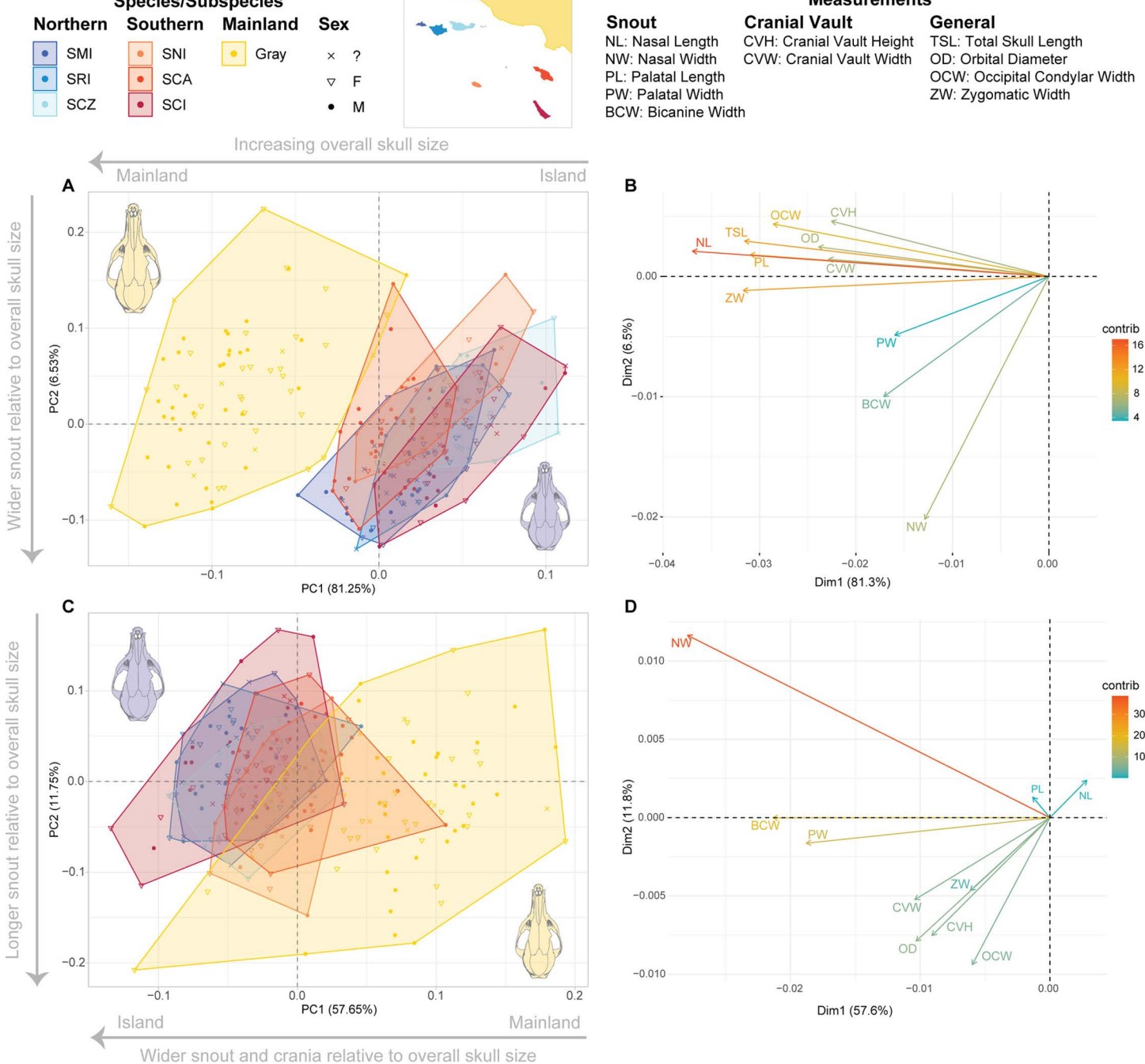

**Fig 6. Principal Component Analysis of gray fox and island fox subspecies examining variations in shape and size driven by linear measurements.** Diagram of measurements shown in Fig 2. PCA and variable contribution of log10 raw linear measurements (contrib) is shown in A and B. PCA and variable contribution of log10 measurements scaled to the total skull length (TSL) is shown in C and D. Skull illustration in A and C represents island fox (purple) and gray fox (yellow). Specific measurements included in the eigenvector loadings are listed above.

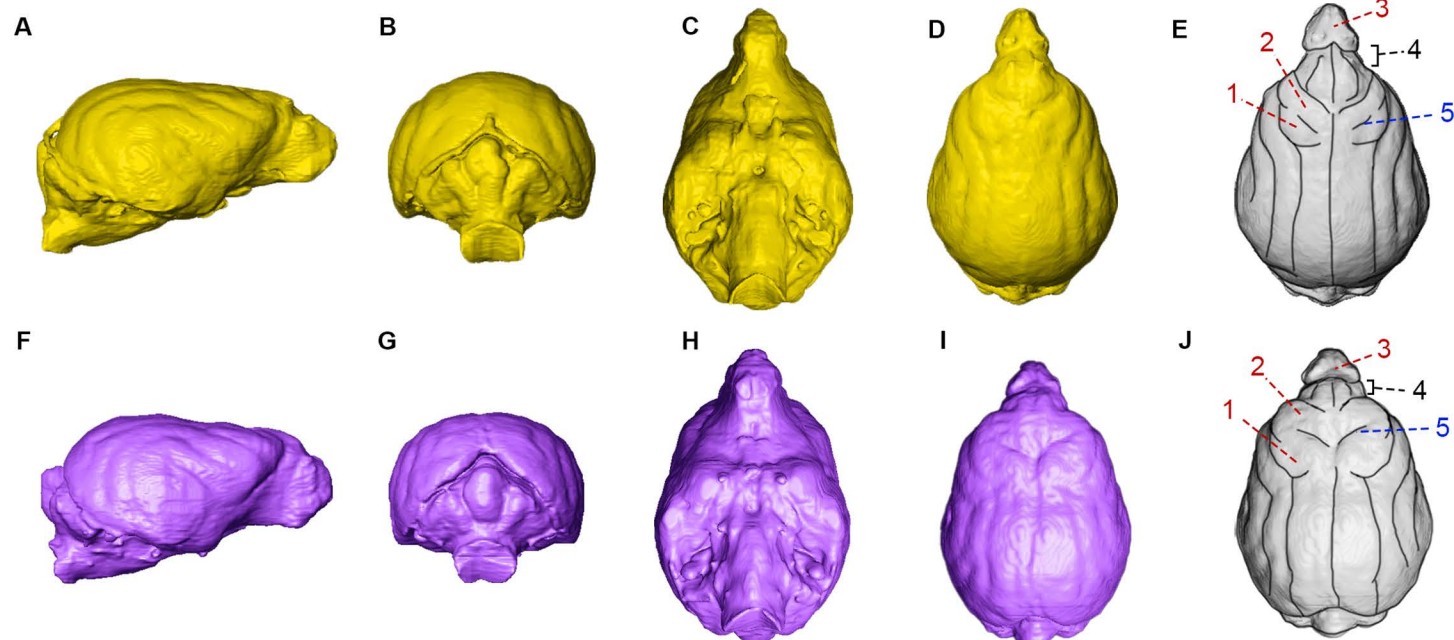

**Fig 7. Brain endocasts of the gray fox (*U. c. californicus*) specimen LACM 87421 (yellow, A-E) and the island fox (*U. l. catalinae*) LACM 75000 (purple, F-J).** Views from L-R: right lateral (A and F), posterior (B and G), ventral (C and H), and dorsal (D and I). Illlustrated dorsal views (E and J) highlight key cerebral structures: postcruciate gyri (1), precruciate gyri (2), olfactory bulb (3), prefrontal area (4), and cruciate sulci (5).

**Table 3. Difference in encephalization quotient (EQ) between island species and their respective mainland relatives.**

| Higher taxon | Island species | Mainland relative | EQ change | BM change (kg) |
|---|---|---|---|---|
| Hominidae[1] | *Homo floresiensis*† | *Homo erectus* (early form)† | −0.47 | −23.0 |
| Bovidae[2] | *Myotragus balearicus*† | *Gallogoral meneghinii*†* | −0.53 | −91 |
| Multituberculata[1] | *Litovoi tholocephalos*† | *Ptilodus montanus*† | −0.57 | +0.1 |
| Elephantidae[3] | *Palaeoloxodon falconeri*† | *Palaeoloxodon antiquus*† | +2.6 | −7787 |
| Hippopotamidae[3] | *Hippopotamus lemerlei*† | *Hippopotamus amphibius* | +0.07 | −1120 |
| Canidae[3] | *Cynotherium sardous*† | *Xenocyon lycaonoides*† | +0.30 | −18.0 |
| Cervidae[3] | *Candiacervus ropalophorus*† | *Dama dama* | +0.11 | −13.7 |
| *Urocyon* | *Urocyon littoralis* (subspecies mean) | *Urocyon cinereoargenteus* (species mean) | +0.09 | −1.65 |
| | *U. l. littoralis* | | +0.09 | −1.59 |
| | *U. l. santarosae* | | +0.08 | −1.61 |
| | *U. l. santacruzae* | | +0.22 | −2.05 |
| | *U. l. dickeyi* | | −0.09 | −1.34 |
| | *U. l. catalinae* | | +0.09 | −1.39 |
| | *U. l. clementae* | | +0.16 | −1.96 |

Values calculated from insular EQ studies ([1]Csiki-Sava et al., 2018; [2]Köhler & Moyà-Solà, 2004; [3]Lyras, 2019) and from group means in this study. Extinct species indicated by with a †. *Brain size data is unavailable for the closer relatives of *Myotragus* (the Miocene archaic goats *Aragoral* and *Norbertia* [42]), so *Gallogoral meneghinii*, its Pleistocene relative, is used here for comparison. As such, the reduction in encephalization may less dramatic when compared to Late Miocene taxa [43].

energetically expensive. Higher EQ may contribute to greater spatial cognition and cognitive processing—larger-brained carnivoran species have been observed to be more successful at colonizing new environments, which may be driven by the fact that their relative brain size has also been found to be predictive of problem-solving success [47,48]. Further, some studies have suggested that a prolonged growth period in carnivorans may provide a mechanic under which a larger brain may develop, without risk of tissue starvation in the face of unpredictable resource access [49,50]. This may be applicable to our findings in *Urocyon*, as the island fox is more likely to reproduce in their second year [51] compared to the gray fox that typically reproduces within their first year [16], which may be indicative of a longer period of early development in the island fox. Encephalization in carnivorans has also been found to be negatively associated with geographic range [48], so the small range of the Channel Islands may be correlated with their moderately larger relative brain size. We may further infer that despite the general resource limitation that has caused the overall size reduction in all island foxes, the island sizes (Table 1) may not be small enough to cause a severe selection pressure to reduce EQ, with the single exception of San Nicolas Island. Of the two smallest islands, San Miguel has less area than San Nicolas, but San Miguel's proximity and prior connection to the larger Santa Rosa Island may have provided a resource buffer not available to San Nicolas. The presence of a pygmy mammoth, *Mammuthus exilis*, on the northern Channel Islands [52] further suggests that total biological productivity is sufficient to support an herbivore hundreds of times larger than island foxes.

## Locomotion and diet

*Urocyon* is also the only canid group that is known to exhibit arboreality [53], which is a cognitively complex trait involving spatial navigation of random and uneven three-dimensional space [54]. Along with the potential for high encephalization leading to successful colonization, the limited resources on the island may drive increased arboreality to obtain foods that are common in island fox diets, including fruits and birds [55]. While gray foxes retain this trait as well, the more abundant mainland food sources may not require such an increase in use as may be characteristic of the islands that possess trees. Only four of the six fox-inhabited islands have trees—complex woodlands found on Santa Rosa, Santa Cruz, Santa Catalina, and limited sparse groves found on San Clemente [56]. As such, the presence of these trees as a dietary resource may be a selective pressure behind the higher encephalization on these islands.

Retained tree-climbing ability and increased reliance on tree-based nutrients may also be associated with the increased complexity of frontal lobe folding that we found in the brain endocasts of the island fox relative to the gray fox; higher complexity in the cruciate sulci and gyri in other mammals has been suggested to be associated with an increased reaction speed and somatosensory processing [57]. The combined compression of the frontal area to increase the depth of the sulci and prominence of the gyri may be a result of the brachycephalic development and subsequent compensation in the island fox compared to the gray fox in order to retain complex motor cognition, one of the key functions of the general prefrontal and frontal regions in canids [58,59]. A similar pattern is exhibited in other canids such as the racoon dog *Nyctereutes procyonoides*, where a shortening of the muzzle was associated with an enlarged frontal lobe and broadening of the proreal gyri [60].

## Competition

Competition may also be a driving factor for two of the islands, Santa Cruz and Santa Rosa, as they are the only locations with a competitor species, the island spotted skunk (*Spilogale gracilis amphiala*) [61]. The two species have likely coexisted on these islands for the majority of their shared history, with the skunks estimated to have arrived ~9000 years ago to the superisland of Santarosae [62]. Although exact spotted skunk population numbers on each island are not known, a study examining genetic diversity of the two populations indicated higher nucleotide diversity on Santa Cruz, which may indicate of higher effective population size [62]. As such, competition for similar dietary items such as insects and mice [63] and den habitats throughout island cohabitation may be a contributing factor for high encephalization of the island fox, particularly on Santa Cruz.

## Limitations for San Nicolas foxes

Of the six island foxes, the San Nicolas fox is the only subspecies with a reduced EQ relative to its mainland relative, though it does not exhibit the greatest body size reduction [12]. The most straight-forward interpretation seems to be the small size of San Nicolas Island, which is one of the smallest Channel Islands (excluding tiny islands beside the big six). If there are low enough resources, the trade-off of allocating energy toward developing neural tissue and cognitive processing may become overwhelmingly negative and energy storage may be prioritized. This may be the case in the relative brain size on San Nicolas Island, which is the least resource-abundant of all six fox-inhabited islands [64]. The landscape lacks trees and is dominated by low shrubs [64], eliminating the need for complex spatial navigation required for climbing. San Nicolas is also the most geographically isolated, both from other islands and from the mainland (Fig 1, Table 1), and has the lowest within-population genetic diversity [11,65–67]. The low relative brain size, as such, may stem from the need to allocate energy towards less energetically expensive survival traits.

## Human assisted dispersal and cohabitation

There are many complexities involved in interpreting our findings in the six subspecies of island fox. In particular, we must consider the cohabitation and relationship between island foxes and humans. Domesticated carnivorans are known to have substantially smaller relative brain sizes than wild ancestral relatives [24]. While the island fox was not what we may think of as domesticated in the modern sense, it has become a prevalent hypodigm in recent years that the foxes were transported to the southern islands by indigenous peoples and lived alongside them in some capacity [11,20,21].

Our findings are relevant to the question of human cohabitation. The overall increased EQ among all subspecies except that from San Nicolas Island strongly hints that cohabitation with human was limited. Whatever the relationship, true domestication of island foxes (as in the mode of domestic dogs) was obviously not practiced by the indigenous people, despite the fact that domestic dogs apparently accompanied the initial peopling of the Americas [68], i.e., living with a top predator was by then widely accepted by humans. We may thus speculate that early dog domestication in the late Pleistocene was mostly for utilitarian reasons (such as hunting assistance and a source of food) rather than companionship, and as such, the island foxes may provide a counter example that small foxes, with apparently little survival utility for humans except as food, were not worth domesticating (sharing human resources with). Any utility of island foxes to humans may then parallel more similarly to benefits of early cat domestication rather than that of domestic dogs. With diets high in insects and deer mice [55], island foxes may have been transported by and lived in proximity to humans as a form of pest control. While not necessarily being fed directly by humans, they may have lived closely to gain access to their readily available natural prey species that may have been drawn to these human settlements, as is a prominent theory in early cat domestication [69]. Whatever the reason indigenous people chose to transport foxes from island to island, they probably did not develop close enough a relationship to cause island foxes to reduce their EQ.

However, our findings show disparity between the three islands in terms of relative brain size despite all likely having been transported by humans. This disparity might be explained by the larger size and topographic complexity of Santa Catalina and San Clemente—following initial transport, their original fox populations may have spent less time interacting with and relying on humans for resources than on San Nicolas.

## Conclusions

This study in conjunction with other findings of increased EQ in island dwarfs provides evidence to support the removal of reduced relative brain size as a component of "Island Syndrome". Our findings indicate that there is more nuance and complexity to body size scaling during island adaptation—the island fox is not simply a smaller version of the mainland gray fox. On average, the island fox presented with a moderately higher EQ than the gray fox and five out of six of the island fox subspecies demonstrate an increase in relative brain size, with only the most isolated and resource-poor island demonstrating relative brain size reduction. Our findings also show that despite being transported by humans and living alongside one another, island foxes do not demonstrate changes consistent with encephalization in domesticated animals and were likely not true domesticates.

Further, this study shows that there is a need for examination of both broader reaching and extant taxa in the study of island brain size, as many of the characteristics that may contribute to the island fox's increased EQ pertain specifically to predator species (hunting, problem-solving, etc.) and not to the grazing herbivorous species that make up many island studies. The island fox is an apex predator on the islands, and though it benefits from the lack of threat of higher-level predators, retention of cognitive abilities is likely paramount to their ability to find prey and obtain food through hunting and scavenging, which may not be the case for species that rely on grazing or other less-cognitively complex means of obtaining nutrients. This higher EQ likely also provides an advantage in these areas over their competitor, the island spotted skunk, on Santa Cruz and Santa Rosa. Further investigation into brain size in other insular carnivores, as such, may provide even more insights into the variation in adaptive mechanisms that occur in these environments.

Overall, from these findings, we submit three key takeaways. First, island foxes were likely not "domesticated" by indigenous peoples, but rather may have been transported and lived in proximity as a means of pest control for human settlements on the islands. Second, reduced encephalization should not be considered a straightforward trait inherent to island adaptation due to nuances in the tradeoffs of energy usage involved in species' resilience in insular environments. Third, despite being an energetically costly tissue, carnivoran brains play crucial roles in their daily functioning and as such, may be selectively advantageous to maintain at a greater relative size even as resources limit overall body size.

## Supporting information

**S1 Fig. Comparisons among gray fox subspecies in body mass (A), total skull length (B), scaled encephalization quotient (C), and scaled brain to body mass ratio (D).** Means in all plots share the same letter, indicating no significant difference by Tukey-test at 5% level of significance.
(TIF)

**S2 Fig. Relative brain size (BBMR) of gray fox and island fox subspecies.** Values scale normalized to zero. Means not sharing any letter are significantly different by Dunnett's T3 test at 5% level of significance.
(TIF)

**S3 Fig. Relative brain size (EQ) between sexes within each geographic group.** EQ values scale normalized to zero. Within group *p*-values for Tukey statistical differences shown in brackets above each pairing.
(TIF)

**S1 Table. Full list of specimens used in this study with raw linear and volumetric measurements.** Source of specimen indicated under column "collection": NHM (Natural History Museum of Los Angeles County) or SB (Santa Barbara Museum of Natural History). Linear measurements in millimeters, volumetric measurements in milliliters (cubic centimeters).
(XLSX)

**S2 Table. Comparison of manual and digital endocranial volumes.** Manual volumes were measured via bead displacement, digital volumes measured using endocast segmentation and surface volumes in Avizo.
(PDF)

**S3 Table. Bonferroni-adjusted *p*-values from post-hoc comparisons from PCA of gray fox and island fox subspecies.** Comparisons above dashed lines indicate results from PCA of log10 raw linear measurements, comparisons below dashed lines indicate results from PCA of log10 normalized measurements.
(PDF)

**S1 Appendix. R code used in this study.**
(PDF)

## Acknowledgments

We thank our collaborators at the Natural History Museum of Los Angeles County (Shannen Robson and Kayce Bell) and at the Santa Barbara Natural History Museum (Jonathan Hoffman and Krista Fahy) for sharing collections and specimens; and the Edmands Lab (Alice Coleman, Jake Denova, Eliza Kirsch, Scott Applebaum) and Kimberly's PhD committee members (Carly Kenkel, Melissa Guzman, Julia Schwartzman, Adam Huttenlocker, and Michael Campbell) for support and feedback.

## Author contributions

**Conceptualization:** Kimberly A. Schoenberger, Xiaoming Wang, Suzanne Edmands.

**Data curation:** Kimberly A. Schoenberger.

**Formal analysis:** Kimberly A. Schoenberger.

**Funding acquisition:** Xiaoming Wang, Suzanne Edmands.

**Investigation:** Kimberly A. Schoenberger.

**Methodology:** Kimberly A. Schoenberger.

**Project administration:** Kimberly A. Schoenberger.

**Resources:** Kimberly A. Schoenberger, Xiaoming Wang, Suzanne Edmands.

**Software:** Kimberly A. Schoenberger.

**Supervision:** Xiaoming Wang, Suzanne Edmands.

**Validation:** Kimberly A. Schoenberger.

**Visualization:** Kimberly A. Schoenberger.

**Writing – original draft:** Kimberly A. Schoenberger.

**Writing – review & editing:** Xiaoming Wang, Suzanne Edmands.

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
