## [Decision Letter · Decision Letter 0]

PONE-D-25-13659Increased brain size of the dwarf Channel Island fox (*Urocyon littoralis* ) challenges “Island Syndrome” and suggests little evidence of domesticationPLOS ONE

Dear Dr. Schoenberger,

Thank you for submitting your manuscript to PLOS ONE. After careful consideration, we feel that it has merit but does not fully meet PLOS ONE’s publication criteria as it currently stands. Therefore, we invite you to submit a revised version of the manuscript that addresses the points raised during the review process.

First clarify the taxonomic scale of study. If subspecies is available for grey fox this should be considered and implemented. Sexual dimorphism should also be tested more explicitly together with the implementation of stats tests for slope between grey and island fox. If available, report the area of each island and run a simple correlation test between average skull length and island area.Provide raw data as appendix or in a repository and implemented literature and other concepts as advised by reviewers.  

We look forward to receiving your revised manuscript.

Kind regards,

Carlo Meloro

Academic Editor

PLOS ONE

2. In your manuscript, please provide additional information regarding the specimens used in your study. Ensure that you have reported human remain specimen numbers and complete repository information, including museum name and geographic location.

For more information on PLOS ONE's requirements for paleontology and archeology research, see https://journals.plos.org/plosone/s/submission-guidelines#loc-paleontology-and-archaeology-research.

“We thank our collaborators at the Natural History Museum of Los Angeles County (Shannen Robson and Kayce Bell) and at the Santa Barbara Natural History Museum (Jonathan Hoffman and Krista Fahy) for sharing collections and specimens; the Edmands Lab (Alice Coleman, Jake Denova, Eliza Kirsch, Scott Applebaum) and Kimberly’s PhD committee members (Carly Kenkel, Melissa Guzman, Julia Schwartzman, Adam Huttenlocker, and Michael Campbell) for support and feedback; and Daniel Muhs for providing island maps. Funding for this project was provided by Dornsife College of Letters, Arts and Sciences at the University of Southern California; the Wrigley Institute and Offield Family Foundation; and the USC Women in Science and Engineering Graduate Fellowship.”

Additional Editor Comments:

The paper reads well and it is nice however my main point of concern is your way of considering U. cinereoargenteus as a single species vs 6 subspecies. In theory, your paper will be much stronger if you could account for subspecies differences also within your grey fox sample. Also you should introduce sexual dimorphism earlier in the paper and test for it. One consequence of island syndrome might also be increase in sexual size dimorphism (see: Raia, P., Barbera, C., & Conte, M. (2003). The fast life of a dwarfed giant. Evolutionary Ecology, 17, 293-312). You have an excellent opportunity to test for it by using simple SSD ratios for each of your island subspecies. As an example see on how to compute this see: Cardini, A., & Elton, S. (2008). Variation in guenon skulls (II): sexual dimorphism. Journal of Human Evolution, 54(5), 638-647.

Below I provide more specific points of concerns for each section that I hope you can find useful to increase clarity in your paper. I think the paper include quite valuable data and information. If you could make raw data available [eg. measurements at least] that will also be good.

Introduction

Line 58-59: add scientific names since this is first time you mention these species

Line 66: perhaps refer to Figure 1 and add some arrows on the map showing the potential route of island colonisation with dates. Add in the map also a mini-map outlining the area within the broader context of North American continent

Mat & Method:

Line 92: since you refer to the island subspecies it is worth providing more details on your sample distribution for U. cinereoargenteus for which there have been recorded about 16 subspecies....did you use only skulls of one population or did you mix specimens from different locations / subspecies....in that case it might be important to outline this. You want to produce a fair comparison across subspecies (variation of subspecies of A vs variation of subspecies of B) otherwise the comparison will be unfair (A might vary more geographically making the false impression of bigger variation compared to subspecies of B).

Line 126: do you mean cranial morphometry? You do not present any geometric morphometric analysis that is also based upon anatomical landmarks but in geometric morphometrics the raw data are coordinates and NOT linear distances between landmarks.

Line 138: nice one....I wonder how does this compare to the Van Valkenburgh equations on Canidae only. It would make more sense to use Canidae specific equation for a more accurate body mass reconstruction

Results

Line 161: because you are talking about scaling I was expecting a test of slope differences and not a test of variance. It might be worth exploring the regression of Skull length (X) vs Brain Size between the two subspecies or different subspecies. This test will address the question: does grey fox and island fox brain size scale in the same way? Also I can see Sex in Figure S1 -this should be in the paper since it is your very first result- but Sex is not mentioned in the Introduction / Mat-Methods...if you got sexual dimorphism you should first test for differences in relative brain size between species and sex using a model "BrainSize~Species+Sex+Species*Sex"....if Species*Sex is significant you might need to do the analyses separated by sex (e.g. compare males with males only). IF you have sufficient subspecies sample size for U. cinereoargenteus than your unit of analysis [also for sex] should be "subspecies" and not species.

Line 162: why did you use Wilcoxon [generally it applies to dependent data that have a time structure [e.g. before and after]...you should use simple t-test or non-parametric Mann-Whitney if your groups are species.

Line 165: Report slopes and intercepts (and 95% CI if necessary) for your models in a table....slopes of the two Urocyon seem parallel to me but without any stats test we will never know that for sure...try and test for difference using ANCOVA and eventually check if slope differ between subspecies (if n within subspecies is big enough).

Line 177: delete "with"

Line 179-181: leave it out for the discussion

Line 191: you cannot talk of shape for plot 5B -it is mainly size, right?-. Different thing is plot 5C. A log transformation of the data might make the plot also a bit better. You can clearly see the issue of species (big yellow convex hull) vs subspecies (convex hulls there are much much smaller)

Line 192-197: you should test for differences using MANOVA/CVA...if test is significant then with post-hoc you can identify the pair of taxa that are more disparate between each other...based on the plot there seems to be no difference between subspecies of island fox so statistically it is more correct to not to consider the populations as separate analytical entities [if they do not differ and you have no subspecies of U.cinereoargenteus it is more correct to merge them into a single group...which is obviously not what you want to do!]

Line 225: if you manage to find subspecies for U. cinereoargenteus perhaps your differences might be even larger...at the moment the only true difference I feel to trust if the one between grey and island fox.

Discussion

Line 269: nice explanation perhaps it is worth talking also about the competitors...do they have many on the island or reduced competition (check for theoretical background: Raia, P., & Meiri, S. (2006). The island rule in large mammals: paleontology meets ecology. Evolution, 60(8), 1731-1742)? does diet change between gray and island fox? any refs on diet?

Line 290: you can check for this in your data testing the association between average skull size and island area for the six island subspecies (van der Geer, A. A., van den Bergh, G. D., Lyras, G. A., Prasetyo, U. W., Due, R. A., Setiyabudi, E., & Drinia, H. (2016). The effect of area and isolation on insular dwarf proboscideans. Journal of Biogeography, 43(8), 1656-1666)

Reviewers' comments:

Reviewer's Responses to Questions

**Comments to the Author**

1. Is the manuscript technically sound, and do the data support the conclusions?

Reviewer #1: Yes

Reviewer #2: Yes

2. Has the statistical analysis been performed appropriately and rigorously? 

Reviewer #1: Yes

Reviewer #2: Yes

3. Have the authors made all data underlying the findings in their manuscript fully available?

Reviewer #1: Yes

Reviewer #2: No

4. Is the manuscript presented in an intelligible fashion and written in standard English?

Reviewer #1: Yes

Reviewer #2: Yes

5. Review Comments to the Author

Reviewer #1: General remark:

One of the most significant aspects of this paper is the size of its sample. Most studies on the brain size of insular mammals rely on fossil specimens, often with very limited sample sizes (sometimes just a single specimen). In contrast, this paper examines 297 specimens, which is a considerable strength that could be emphasized more.

I also have one suggestion that the authors may find interesting. It is related to the brain anatomy.

The authors noticed that the island fox has a reduced length in the prefrontal area. I believe this is a consequence of its shorter rostrum. In general, canids that have relatively short faces have relatively high and massive frontal brain lobes. A somewhat similar case to Urocyon is the living raccoon dog Nyctereutes procyonoides. That species has a shorter muzzle than its Pliocene relatives. The muzzle shortening led to a shortening of the proreal gyrus length (Lyras 2009).

On Table 2, I have three minor comments:

1. Csiki-Sava et al. (2018) compare Homo floresienceis with an early form of Homo erectus ‘Homo erectus (early form)’. I suggest they add a similar parenthesis in their table, as brain sizes differ significantly between early and late forms of Homo erectus.

2. They list Gallogoral meneghinii as the ancestor of Myotragus balearicus. Actually that is not true, and neither the reference they cite (Köhler et al., 2004), says so. Gallogoral meneghinii is a Pleistocene relative of Myotragus. Myotragus is phylogenetically related to the Miocene archaic goats Aragoral and Norbertia (Rozzi, 2013). Unfortunately, their brain sizes are unknown. However, we do know that the brains Late Miocene bovids are in general smaller than those of modern bovids (Liakopoulou et al., 2024). That makes the reduction of brain size of Myotragus less dramatic than that reported by Köhler et al. (2004). To keep things simple, I suggest keeping Gallogoral meneghinii in the table but adding a footnote clarifying that it is a Pleistocene relative of Myotragus. Additionally, the footnote could mention that the reduction in encephalization appears less dramatic when compared to Late Miocene taxa (Liakopoulou et al., 2024).

3. They list Cervus elaphus as the ancestor of Candiacervus ropalophorus. It has been suggested that the fallow deer is the living relative of Candiacervus (van der Geer, 2018). I am sure that the replacement of Cervus elaphus with Dama dama will not significantly alter the values in the table.

Citations:

Csiki-Sava Z, Vremir M, Meng J, Brusatte SL, Norell MA. 2018. Dome-headed, small-brained island mammal from the Late Cretaceous of Romania. Proc Natl Acad Sci U S A. 115(19):4857–4862.

Köhler M, Moyà-Solà S. 2004. Reduction of brain and sense organs in the fossil insular bovid Myotragus. Brain Behav Evol. 63(3):125–140.

Liakopoulou D, Roussiakis S, Lyras G. 2024. The brain of Myotragus balearicus, an insular bovid from the Balearics. Hist. Biol. 1–8.

Lyras G.A. 2009. The evolution of the brain in Canidae (Mammalia: Carnivora). Scripta Geologica 139: 1-93.

Rozzi R. 2013. Palaeobiogeography and evolution of insular bovids: ecogeographic patterns of body mass variation and morphological changes. Unpublished Ph.D. Thesis. Sapinza Università di Roma, Dipartimento di Scienze sella Terra.

van der Geer A.A.E. 2018. Uniformity in variety: Antler morphology and evolution in a predator-free environment. Palaeontol Electron. 25.2.a23.

Reviewer #2: This is a thorough study addressing brain size and some external anatomical features of the brain of the Channel Island foxes of several islands respective to their mainland congener, the gray fox. It is an excellent study, well structured and easy to read. I have no suggestions for improvement, but have a very few remarks. First of all, I selected no for data availability, because this is not clear to me: are the CT-scans available for external brain morphology? Then, in the Introduction, line 51, what means 'simplified locomotion'? I'm not aware of such a feature in island endemic mammals, but my guess is that here reduced dispersal abilities are meant, such as loss of running in vertebrates, of flight in birds. In any case, 'simplified' needs to be explained here. Same with coloration, it's rather dull than simplified, and applies only (?) birds (mammals can e.g. develop spotted patterns, white-tipped tail etc., which is not 'simplified coloration'. Further, in line 60 no reference is given for the body size reduction in the island foxes (could be Lyras et al. 2010 JoB; but perhaps there is a more recent reference). Then, page 4 lines 82-83 the definition of endemic includes also mainlands, so the addition 'island' or 'insular' is needed here. There are many more living endemic canids, but if you excluse the mainlands, then the statement is almost true (there is also the Cozumel island fox). Discussion page 13 line 239 reduced predatory pressure does not apply mostly to small vertebrates; I'd argue the opposite. Elephants evolve dwarfism when no predators are around, and they are not small. Also, for rodents, predatory pressure is not lower on islands, but shifted to birds only (which can grow gigantic). Page 15 last paragraph, this link with increased arboreality is very interesting, and looks like a strong point. So I was wondering, how is the situation on San Nicolas? Page 16 line 316 I don't agree that the first domestic dogs in the late Pleistocene were mostly hunting assistants; archaeological evidence suggests rather that they were primarily food items (as with horses). Lastly, in the bibliography, a few titles are erroneously given in caps (e.g. 11, 27, 36

6. PLOS authors have the option to publish the peer review history of their article (what does this mean? ). If published, this will include your full peer review and any attached files.

**Do you want your identity to be public for this peer review?** For information about this choice, including consent withdrawal, please see our Privacy Policy .

Reviewer #1: No

Reviewer #2: No

---

## [Author Response · Author response to Decision Letter 1]

11 May 2025

Dear Carlo Meloro and reviewers,

Thank you for your comments and advice on revisions for our research article, “Increased brain size of the dwarf Channel Island fox (Urocyon littoralis) challenges ‘Island Syndrome’ and suggests little evidence of domestication”. Our responses are shown below and match those found in the attached PDF response to reviewers letter.

Editor comments:

Intro notes:

1. “First clarify the taxonomic scale of study. If subspecies is available for grey fox this should be considered and implemented.”

a. We have updated the manuscript to explicitly state the taxonomic scale of the study, outlining the gray fox subspecies used and the geographic distribution of mainland specimens (Lines 80-90, 134-135, 144-154, 271-286, Fig 1A, Fig 6, S1 Fig).

2. “Sexual dimorphism should also be tested more explicitly together with the implementation of stats tests for slope between grey and island fox.”

a. Sexual dimorphism has been previously examined in the island and gray fox by multiple other studies and has now been outlined explicitly in this manuscript (Lines 71-73), with additional comments about the EQ sex differences from our findings (Lines 31, 261-262, S2 Fig).

3. “If available, report the area of each island and run a simple correlation test between average skull length and island area.”

a. The area of each island is noted in Table 1, and body size among islands has been added from prior studies (Lines 65-68), with little correlation between body size and land area in the island fox.

4. “Provide raw data as appendix or in a repository and implemented literature and other concepts as advised by reviewers.”

a. Raw data of endocasts has been added to a Morphosource repository.

Journal Requirements:

1. “Please ensure that your manuscript meets PLOS ONE's style requirements, including those for file naming.”

a. Manuscript has been updated to match style and file naming requirements for PLOS One.

2. “In your manuscript, please provide additional information regarding the specimens used in your study. Ensure that you have reported human remain specimen numbers and complete repository information, including museum name and geographic location.”

a. Permits and specimen info have been added to comply with PLOS One requirements (Lines 130-132).

3. “We note that you have provided funding information that is not currently declared in your Funding Statement. However, funding information should not appear in the Acknowledgments section or other areas of your manuscript.”

a. No funding has been provided for this project specifically, comments regarding funding in the acknowledgements were regarding general funding and support for Kimberly’s PhD. Phrasing in Acknowledgments has been changed to refer to only the support provided (Lines 480-483), and funding statement declaration has been updated.

4. “We note that Figure 1 in your submission contain [map/satellite] images which may be copyrighted.”

a. Figure 1 has been modified to use public datasets for the main body which are cited explicitly in the figure caption, and copyright permission has been obtained from Cambridge University Press, with proof of permission uploaded along with this revised submission.

5. “Please review your reference list to ensure that it is complete and correct.”

a. Reference list has been updated to accommodate requested revisions, with some changes to the order of the original references as edits were made. The following references added:

i. Meiri S, Simberloff D, Dayan T. Insular carnivore biogeography: Island area and mammalian optimal body size. American Naturalist. 2005;165(4):505–14. doi: 10.1086/428297

ii. Schutz H, Polly PD, Krieger JD, Guralnick RP. Differential sexual dimorphism: size and shape in the cranium and pelvis of grey foxes (Urocyon). Biological Journal of the Linnean Society. 2009;96(2):339–53. doi: 10.1111/j.1095-8312.2008.01132.x

iii. Raia P, Barbera C, Conte M. The fast life of a dwarfed giant. Evol Ecol. 2003;17(3):293–312. doi: 10.1023/A:1025577414005

iv. Reding DM, Castañeda-Rico S, Shirazi S, Hofman CA, Cancellare IA, Lance SL, et al. Mitochondrial Genomes of the United States Distribution of Gray Fox (Urocyon cinereoargenteus) Reveal a Major Phylogeographic Break at the Great Plains Suture Zone. Front Ecol Evol. 2021;9. doi: 10.3389/fevo.2021.666800

v. Hofman CA, Rick TC, Hawkins MTR, Funk WC, Ralls K, Boser CL, et al. Mitochondrial Genomes Suggest Rapid Evolution of Dwarf California Channel Islands Foxes (Urocyon littoralis). PLoS One. 2015;10(2):e0118240. doi: 10.1371/journal.pone.0118240

vi. Gompper ME, Petrites AE, Lyman RL. Cozumel Island fox (Urocyon sp.) dwarfism and possible divergence history based on subfossil bones. J Zool. 2006;270(1):72–7. doi: 10.1111/j.1469-7998.2006.00119.x

vii. Collins PW. Origin and Differentiation of the Island Fox: A Study of Evolution in Insular Populations [Master of Arts]. Santa Barbara: University of California; 1982.

viii. Massicotte P, South A. rnaturalearth: World Map Data from Natural Earth. CRAN: Contributed Packages. 2017. doi: 10.32614/CRAN.package.rnaturalearth

ix. National Geodetic Survey. NOAA National Shoreline Data Explorer [Shapefile]. https://nsde.ngs.noaa.gov/. National Oceanic and Atmospheric Administration; [accessed 30 Apr 2025] Available from: https://nsde.ngs.noaa.gov/

x. McGee S. Continental Divide-Pacific/Atlantic [Shapefile]. ArcGIS Hub. U.S. Fish & Wildlife Service; 2023. [accessed 29 Apr 2025] Available from: https://hub.arcgis.com/datasets/fws::continental-divide-pacific-atlantic/about

xi. Rozzi R. Palaeobiogeography and evolution of insular bovids: ecogeographic patterns of body mass variation and morphological changes (Unpublished Ph.D Thesis). Sapinza Università di Roma; 2013.

xii. Liakopoulou D, Roussiakis S, Lyras G. The brain of Myotragus balearicus , an insular bovid from the Balearics. Hist Biol. 2024;1–8. doi: 10.1080/08912963.2024.2406962

xiii. Lyras G. The evolution of the brain in Canidae (Mammalia: Carnivora). Scr Geol. 2009;139.

xiv. Floyd CH, Van Vuren DH, Crooks KR, Jones KL, Garcelon DK, Belfiore NM, et al. Genetic differentiation of island spotted skunks, Spilogale gracilis amphiala. J Mammal. 2011;92(1):148–58. doi: 10.1644/09-MAMM-A-204.1

xv. Bolas EC, Quinn CB, Van Vuren DH, Lee A, Vanderzwan SL, Floyd CH, et al. Pattern and timing of mitochondrial divergence of island spotted skunks on the California Channel Islands. J Mammal. 2022;103(2):231–42. doi: 10.1093/jmammal/gyac008

xvi. Pasciullo Boychuck S, Brenner LJ, Gagorik CN, Schamel JT, Baker S, Tran E, et al. The gut microbiomes of Channel Island foxes and island spotted skunks exhibit fine‐scale differentiation across host species and island populations. Ecol Evol. 2024;14(2). doi: 10.1002/ece3.11017

Additional Editor Comments:

General:

1. “The paper reads well and it is nice however my main point of concern is your way of considering U. cinereoargenteus as a single species vs 6 subspecies. In theory, your paper will be much stronger if you could account for subspecies differences also within your grey fox sample. Also you should introduce sexual dimorphism earlier in the paper and test for it.”

a. As mentioned above, subspecies differences within the gray fox sample have been explicitly addressed and tested throughout the manuscript (see line numbers above). Sexual size dimorphism has also been addressed in the paper, including reference to the suggested Raia et al. (2003) paper (Lines 71-75). As SSD has already been largely previously examined in other papers, I refer to those studies and thus did not compute other SSD ratios explicitly with my dataset besides those regarding EQ (S2 Fig).

2. “I think the paper include quite valuable data and information. If you could make raw data available [eg. measurements at least] that will also be good.”

a. Raw data is available in the supplement as S1 Table containing all linear and volumetric measurements for specimens, with collection information and catalog numbers. I have updated the supplement title to explicitly state that it contains the measurement info along with the specimen list.

Line comments:

Introduction

1. “Line 58-59: add scientific names since this is first time you mention these species”

a. Agreed, corrected (Lines 62-64).

2. “Line 66: perhaps refer to Figure 1 and add some arrows on the map showing the potential route of island colonization with dates. Add in the map also a mini-map outlining the area within the broader context of North American continent”

a. Agreed, updated with above comments as well as including a map of the gray fox specimen areas in the context of North America (Fig 1).

Material and Methods

3. “Line 92: since you refer to the island subspecies it is worth providing more details on your sample distribution for U. cinereoargenteus for which there have been recorded about 16 subspecies....”

a. Details for gray fox specimens have been updated here and throughout the paper (see above comments).

4. “Line 126: do you mean cranial morphometry? You do not present any geometric morphometric analysis that is also based upon anatomical landmarks but in geometric morphometrics the raw data are coordinates and NOT linear distances between landmarks.”

a. Yes, corrected (Line 183).

5. “Line 138: nice one....I wonder how does this compare to the Van Valkenburgh equations on Canidae only. It would make more sense to use Canidae specific equation for a more accurate body mass reconstruction”

a. Equations from below reference (Van Valkenburgh, 1990) use four measurements (head-body length, skull length, occiput-orbit length, and lower first-molar length). The only measurement that we were able to test for comparison without collection additional measurements from all specimens was the skull length, which we tested with Van Valkenburgh’s equation to calculate body mass. This resulted in substantial overestimates of known fox body mass. Mean body mass for all islands using Van Valkenburgh equations was well above 3 kg, which is higher than known body mass ranges for island foxes (1.2-2.7 kg). To contrast, averages calculated using our original method of Engelman’s regression of the occipital condylar width (OCW) fell within the known range (Island means ranged between 1.98-2.64 kg). As such, we did not proceed with using the suggested equation.

i. Van Valkenburgh, B. 1990. Skeletal and dental predictors of body mass in carnivores. In: J. Damuth and B.J. MacFadden (eds.), Body Size in Mammalian Paleobiology, 1–11. Cambridge University Press, Cambridge.

Results

6. “Line 161: because you are talking about scaling I was expecting a test of slope differences and not a test of variance. It might be worth exploring the regression of Skull length (X) vs Brain Size between the two subspecies or different subspecies. This test will address the question: does grey fox and island fox brain size scale in the same way? Also I can see Sex in Figure S1 -this should be in the paper since it is your very first result- but Sex is not mentioned in the Introduction / Mat-Methods...if you got sexual dimorphism you should first test for differences in relative brain size between species and sex…”

a. Agreed, added info for testing slope differences in both the original plot (Body Mass vs Brain Size) and new additional plot (Skull Length vs Brain Size). Info can be found in Fig 4 and Table 2.

b. Figure S1 has been moved to the main manuscript as Fig 3. Sexual dimorphism has been addressed in the intro (Lines 71-73) and in S2 Fig.

7. “Line 162: why did you use Wilcoxon [generally it applies to dependent data that have a time structure [e.g. before and after]...you should use simple t-test or non-parametric Mann-Whitney if your groups are species.”

a. This has been corrected, using non-parametric Mann-Whitney (Line 230).

8. “Line 165: Report slopes and intercepts (and 95% CI if necessary) for your models in a table....slopes of the two Urocyon seem parallel to me but without any stats test we will never know that for sure...try and test for difference using ANCOVA and eventually check if slope differ between subspecies (if n within subspecies is big enough).”

a. Slopes and intercepts of each species with tests for significant differences have been added in Table 2.

9. “Line 177: delete "with"”

a. Agreed, deleted.

10. “Line 179-181: leave it out for the discussion”

a. Agreed, removed.

11. “Line 191: you cannot talk of shape for plot 5B -it is mainly size, right?-. Different thing is plot 5C. A log transformation of the data might make the plot also a bit better. You can clearly see the issue of species (big yellow convex hull) vs subspecies (convex hulls there are much much smaller)”

a. Agreed, results have been updated to clarify that differences in raw data are driven by size (Lines 281-286). PCA has been rerun with gray fox subspecies included (Fig 6).

12. “Line 192-197: you should test for differences using MANOVA/CVA...if test is significant then with post-hoc you can identify the pair of taxa that are more disparate between each other...based on the plot there seems to be no difference between subspecies of island fox so statistically it is more correct to not to consider the populations as separate analytical entities…”

a. Agreed, information about additional statistical tests has been added to the manuscript under the methods and results sections (Lines 220-225, 268-281).

13. “Line 225: if you manage to find subspecies for U. cinereoargenteus perhaps your differences might be even larger...at the moment the only true difference I feel to trust if the one between grey and island fox.”

a. Significance tests were run for species-species comparisons as well as subspecies-subspecies comparisons. Species showed substantial significant differences, which was consistent with the subspecies-subspecies pairings and noted in the results section (Lines 268-281).

Discussion

14. “Line 269: nice explanation perhaps it is worth talking also about the competitors...do they have many on the island or reduced competition (check for theoretical background: Raia, P., & Meiri, S. (2006). The island rule in large mammals: paleontology meets ecology. Evolution, 60(8), 1731-1742)? does diet change between gray and island fox? any refs on diet?”

a. Agreed, info about competitors (only one is known (island spotted skunk) and only on two of the islands) has been added into the discussion section (Lines 384-394).

15. “Line 290: you can check for this in your data testing the association between average skull size and island area for the six island subspecies (van der Geer, A. A., van den Bergh, G. D., Lyras, G. A., Prasetyo, U. W., Due, R. A., Setiyabudi, E., & Drinia, H. (2016). The effect of area and isolation on insular dwarf proboscideans. Journal of Biogeography, 43(8), 1656-1666)”

a. This info has been added in the introduction (Lines 65-68), citing other studies that already show lack of correlation between island fox subspecies body size and island area.

Reviewer comments:

Reviewer #1:

1. “One of the most significant aspects of this paper is the size of its sample. Most studies on the brain size of insular mammals rely on fossil specimens, often with very limited sample sizes (sometimes just a single specimen). In contrast, this paper examines 297 specimens, which is a considerable strength that could be emphasized more.”

a. Agreed, added to the materials section (Lines 137-139).

2. “The authors noticed that the island fox has a reduced length in the prefrontal area. I believe this is a consequence of its shorter rostrum. In general, canids that have relatively short faces have relatively high and massive frontal brain lobes. A somewhat similar case to Urocyon is the living raccoon dog Nyctereutes procyonoides. That species has a shorter muzzle than its Pliocene relatives. The muzzle shortening led to a shortening of the proreal gyrus length (Lyras 2009).”

a. Agreed, added to the discussion section (Lines 381-383).

3. “On Table 2, I have three minor comments: 1. Csiki-Sava et al. (2018) compare Homo floresienceis with an early form of Homo erectus ‘Homo erectus (early form)’. I suggest they add a similar parenthesis in their table, as brain sizes differ significantly between earl

---

## [Decision Letter · Decision Letter 1]

PONE-D-25-13659R1Increased brain size of the dwarf Channel Island fox (*Urocyon littoralis* ) challenges “Island Syndrome” and suggests little evidence of domesticationPLOS ONE

Dear Dr. Schoenberger,

Thank you for submitting your manuscript to PLOS ONE. After careful consideration, we feel that it has merit but does not fully meet PLOS ONE’s publication criteria as it currently stands. Therefore, we invite you to submit a revised version of the manuscript that addresses the points raised during the review process.

Double check the use of ANOVA and MANOVA providing more appropriate results. Use covariance matrix for the PCA and make sure you exclude the juveniles from the sample.See the analytical report for advise on how to perform MANOVA and pairwise testing.  

We look forward to receiving your revised manuscript.

Kind regards,

Carlo Meloro

Academic Editor

PLOS ONE

Journal Requirements:

**Additional Editor Comments:**

The paper is almost ready, I just found few minor issues that you can hopefully be implemented very quickly before official acceptance.

Line 193: "Carnivora" upper case

Line 210: It is ANCOVA and NOT ANOVA. Modify as: "We then used ANCOVA, including interaction terms, to test whether slopes and intercepts of these scaling relationships differed significantly between species".

Line 216: I do not think you used "Welch's ANOVA". This is used when variances are not equal, in that case the post doc should be Dunnett's T3.

For an example on how to report this correctly see Meloro (2011, JVP: https://doi.org/10.1080/02724634.2011.550357). You first do a one-way ANOVA followed by a LEvene's test (LEvene's will tell you if variance is homogenous -p > 0.05- or not). If Levene's is NOT significant, you can use Tukey post doc -that will satisfy the assumption of equal variances.

Line 223-224: it is ok to perform MANOVA only on PC1 and PC2 although you might try to cover at least 95% of total variance (e.g. first 5 PCs) and provide results as expressed by Wilk's lambda and Pillai Trace.

Your PCA was probably based on Correlation matrix...you should use Covariance matrix if your input variables are log transformed measurements. See the attach doc "Urocyon_tests". I selected only Adult specimens and provided some examples on analyses run using PAST and SPSS. It is fine to combine all U. cinereoargenteus subspecies but still you need to implement MANOVA more appropriately. ANOVA that follows MANOVA is not appropriate, so please use PAST or try to use simple script in R to run Hotelling's pairwise test after MANOVA. An example is below:

manova_model <- manova(cbind(PC1, PC2, PC3) ~ species

Assuming that your selected PCs are in PC_data

For pairwise comparisons use the library (pairwiseAdonis)

result <- pairwise.adonis(PC_data, factors = species, sim.method = "euclidean", p.adjust.m = "bonferroni")

Report pairwise differences after MANOVA not based on each single PC.

Please clarify if you removed the Juveniles from the sample, I think they should be removed.

Figure 4: It is not clear to what line the R2 and p value refers to. Report eventually the general equation and the R2 next to the general regression line it refers to or use a legend.

You can check the example below:

https://link.springer.com/article/10.1007/s10914-020-09513-w

Only the general equation and the R2 which will be relevant

Figure 5 A and B are identical...perhaps put one or the other in the Appendix

Figure 6: you got one outlier for Fig 6A and 6C. Check carefully, if you remove this individual and PC loadings change that must be removed from the analyses, if not you can keep it. Maybe this occurs when using correlation matrix. See the attached example using the covariance matrix.

I noted from your TAble that you also measured Juveniles...they should be excluded.

Reviewers' comments:

Reviewer's Responses to Questions

**Comments to the Author**

1. If the authors have adequately addressed your comments raised in a previous round of review and you feel that this manuscript is now acceptable for publication, you may indicate that here to bypass the “Comments to the Author” section, enter your conflict of interest statement in the “Confidential to Editor” section, and submit your "Accept" recommendation.

Reviewer #1: All comments have been addressed

Reviewer #2: All comments have been addressed

2. Is the manuscript technically sound, and do the data support the conclusions?

Reviewer #1: Yes

Reviewer #2: Yes

3. Has the statistical analysis been performed appropriately and rigorously? 

Reviewer #1: Yes

Reviewer #2: Yes

4. Have the authors made all data underlying the findings in their manuscript fully available?

Reviewer #1: Yes

Reviewer #2: Yes

5. Is the manuscript presented in an intelligible fashion and written in standard English?

Reviewer #1: Yes

Reviewer #2: Yes

6. Review Comments to the Author

Reviewer #1: (No Response)

Reviewer #2: (No Response)

7. PLOS authors have the option to publish the peer review history of their article (what does this mean? ). If published, this will include your full peer review and any attached files.

**Do you want your identity to be public for this peer review?** For information about this choice, including consent withdrawal, please see our Privacy Policy .

Reviewer #1: No

Reviewer #2: No

---

## [Author Response · Author response to Decision Letter 2]

3 Jul 2025

Dear Carlo Meloro and reviewers,

Thank you again for your comments and advice on revisions for our research article, “Increased brain size of the dwarf Channel Island fox (Urocyon littoralis) challenges ‘Island Syndrome’ and suggests little evidence of domestication”. Our responses are

shown below and match those found in the attached PDF response to reviewers letter.

Note: line numbers below refer to corrections shown in the clean manuscript file. Content is the same in both the clean and markup files, but Microsoft Word has known inconsistencies in line numbering while using “track changes”, so there are some discrepancies in the line numbering between the files despite the writeups being identical.

Editor comments:

Thank you for providing examples of the plot types and calculations you were looking for in this revision. I would like to note that in the example you provided, it appears you also included the “ECV” in the calculations for the PCA and loading values. I have updated the statistical tests using PAST and your other recommendations, but please note that the values will be slightly different from the ones in your example as they do not include ECV due to it being a volumetric measure. They will also differ due to removal of an outlier and clustering of gray fox into one group.

Additionally, copyrighted content has been removed from Figure 1 and replaced with a map generated using only datasets compliant with CC BY 4.0 license. Figure caption has been updated (Lines 161-162).

Line comments:

1. Line 193: "Carnivora" upper case

a. Corrected.

2. Line 210: It is ANCOVA and NOT ANOVA. Modify as: "We then used ANCOVA, including interaction terms, to test whether slopes and intercepts of these scaling relationships differed significantly between species".

a. Corrected (Lines 212-213).

3. Line 216: I do not think you used "Welch's ANOVA". This is used when variances are not equal, in that case the post hoc should be Dunnett's T3.

a. Welch’s ANOVA was initially used due to unequal variance, but revisions have been made per suggestions. Statistical tests used Levene’s test to test for equal variance. Variance was not equal, so proceeded to use one-way ANOVA to assess for statistical differences among group means. Post hoc tests implemented Dunnett’s T3 as suggested. (Lines 214-221).

4. Line 223-224: it is ok to perform MANOVA only on PC1 and PC2 although you might try to cover at least 95% of total variance (e.g. first 5 PCs) and provide results as expressed by Wilk's lambda and Pillai Trace. Your PCA was probably based on Correlation matrix...you should use Covariance matrix if your input variables are log transformed measurements. See the attach doc "Urocyon_tests". I selected only Adult specimens and provided some examples on analyses run using PAST and SPSS. It is fine to combine all U. cinereoargenteus subspecies but still you need to implement MANOVA more appropriately. ANOVA that follows MANOVA is not appropriate, so please use PAST or try to use simple script in R to run Hotelling's pairwise test after MANOVA.

a. Thank you for providing an example of statistical tests! We ended up using PAST for the revised PCA using the variance-covariance matrix, clustering the gray fox as one group. Description of methods outlined in Lines 222-231, and results in Lines 273-289. Ease of use in PAST allowed us to use all principal components for MANOVA and post hoc tests. Statistical differences shown in newly added Table 3 in the manuscript (Line 299). PCA plots generated in R were updated to use the variance-covariance matrix and were checked against plots generated in PAST to ensure consistency. Use of R for plots was chosen to keep figure designs consistent throughout the paper. Figure 6 has been updated to reflect the above changes (Line 291).

5. Report pairwise differences after MANOVA not based on each single PC.

a. Addressed above, see new Table 3 (Line 299).

6. Please clarify if you removed the Juveniles from the sample, I think they should be removed.

a. Juveniles were not included in the sample, added note to state this explicitly (Lines 223-224).

7. Figure 4: It is not clear to what line the R2 and p value refers to. Report eventually the general equation and the R2 next to the general regression line it refers to or use a legend.

a. Agreed, Figure 4 has been updated to include a clearer legend and addition of general equations for each regression. Figure caption updates are on Lines 250-254.

8. Figure 5 A and B are identical...perhaps put one or the other in the Appendix

a. Figure 5A has been left in the main text (Lines 260-262, 269), 5B moved to the appendix S2 fig (Lines 262-264).

9. Figure 6: you got one outlier for Fig 6A and 6C… I noted from your table that you also measured Juveniles...they should be excluded.

a. Thank you, the outlier was identified and removed for PCA revisions and noted in the text (Lines 229-230). Juveniles were not included in the original PCA, but this has now been outlined explicitly in the text (Lines 223-224).

Additional changes to be noted:

1. Figures and supplement:

a. Figs 2-3, Fig 7, S1 Fig, S1 Table, and S2 Table are the same as previous submission with no changes.

b. Fig 1 has been updated to replace copyrighted content with CC BY compliant data.

c. Fig 4 has been updated to include general regression equation and added clarity.

d. Fig 5 has been updated to only include Fig 5A from previous submission, 5B moved to supplement.

e. Fig 6 has been updated with var-covar PCA plots and clustering gray fox subspecies into one group.

f. S1 Appendix has been updated to reflect new code.

g. S2 Fig is the new supplementary figure, previously Fig 5B.

h. S3 Fig is the previous S2 Fig.

2. DOIs have been generated for supplemental Morphosource files, which have now been included under the Data Availability Statement on Lines 499-501.

3. Two references were added for NOAA bathymetry data (Line 593) and PAST Software (Line 610):

a. NOAA National Centers for Environmental Information. ETOPO 2022 15 Arc-Second Global Relief Model. 2022. doi:10.25921/fd45-gt74

b. Hammer Ø, Harper DAT, Ryan PD. PAST: Paleontological statistics software package for education and data analysis. Palaeontologia Electronica; 2001.

Changes as noted above are reflected in the revised manuscript. Thank you again for your time and we look forward to hearing from you.

Sincerely,

Kimberly A. Schoenberger

PhD Candidate, University of Southern California

Graduate Student in Residence, Natural History Museum of Los Angeles County

kschoenb@usc.edu

---

## [Editor Report · Decision Letter 2]

Increased brain size of the dwarf Channel Island fox (*Urocyon littoralis* ) challenges “Island Syndrome” and suggests little evidence of domestication

PONE-D-25-13659R2

Dear Dr. Schoenberger,

We’re pleased to inform you that your manuscript has been judged scientifically suitable for publication and will be formally accepted for publication once it meets all outstanding technical requirements.

Kind regards,

Carlo Meloro

Academic Editor

PLOS ONE

Additional Editor Comments (optional):

You did a good job, I just spotted few things you can adjust during production process:

line 274: representing 87.78% of variance for log-raw measurements, and 69.40% of variance for log-normalized measurements - something not right here because if one PC is 87.78% the second one cannot be 69.40%. You should write PC1 explains 82% of variance and PC2 6%

line 286: "most substantial differences by tens of orders of magnitude greater were found between" remember the P values does not give you the idea of magnitude differences so remove this...you just got highly significant differences.....be aware that this using all variables...if you re-try using only the PCs that explain 95% of variance (e.g. the first five...) you will see few non-significant differences. That might be more worth reporting, this is because MANOVA sometimes overfit differences. Your smallest group should have at least double the number of your variables. In your case if you have 11 variables the smallest group should have at least 22 cases for solid interpretation of MANOVA.

From this perspective the table in the paper is probably not that important -everything is significant- and it can go in the Appendix if you want.

Line 499: "DOIs: 10.17602/M2/M740219 (island fox 500 endocast), 10.17602/M2/M740195 (gray fox endocast), 10.17602/M2/M740169 (island fox raw

501 CT data), 10.17602/M2/M740136 (gray fox raw CT data)". I think these are ID used by morphosource ...if you copy and paste in a browser it does not open your scan so just double check.
---

## [Editor Report · Acceptance letter]

PONE-D-25-13659R2

PLOS ONE

Dear Dr. Schoenberger,

I'm pleased to inform you that your manuscript has been deemed suitable for publication in PLOS ONE. Congratulations! Your manuscript is now being handed over to our production team.

Kind regards,

on behalf of

Dr. Carlo Meloro

Academic Editor

PLOS ONE